

# Technical note: the enhancement limit of coagulation scavenging of small charged particles

Naser G. A. Mahfouz[1,2] and Neil M. Donahue[1,2,3,4]

[1]Center for Atmospheric Particle Studies, Carnegie Mellon University, Pittsburgh, PA, USA
[2]Department of Chemical Engineering, Carnegie Mellon University, Pittsburgh, PA, USA
[3]Department of Chemistry, Carnegie Mellon University, Pittsburgh, PA, USA
[4]Department of Engineering and Public Policy, Carnegie Mellon University, Pittsburgh, PA, USA

**Correspondence:** Neil M. Donahue (nmd@andrew.cmu.edu)

**Abstract.** We show that the limit of the enhancement of coagulation scavenging of charged particles is 2 (doubled compared to the neutral case). Because the particle survival probability decreases exponentially as the coagulation sink increases, all else being equal, the doubling of the coagulation sink can amount to a dramatic drop in survival probability—squaring the survival probability, $p^2$, where $p \leq 1$ is the survival probability in the neutral case. Thus, it is imperative to consider this counterbalanc-

ing effect when studying ion-induced new-particle formation and ion-enhanced new-particle growth in the atmosphere.

## 1   Introduction

There are many situations where we care about total particle number, and especially total particle number above some critical size. An example is cloud activation, where the total number of cloud condensation nuclei is often estimated to be the total number of particles with diameters $d_p \geq 50$ nm ($\mathcal{N}_{50}$) or perhaps $d_p \geq 100$ nm ($\mathcal{N}_{100}$), depending on updraft velocity (Rosen-

feld et al., 2019; Pierce and Adams, 2007; Pierce et al., 2012; Gordon et al., 2017; Lee et al., 2019). When particle formation (nucleation) or emission is dominated by much smaller particles, we thus care not only about the formation or emission rate but also the survival probability of those new particles as they grow to the critical size. That survival probability decreases exponentially as the coagulation sink increases (Kulmala et al., 2017; Kerminen and Kulmala, 2002; Lehtinen et al., 2007; Li and McMurry, 2018). The counterbalancing role of growth rate and coagulation sink is well known, but the role of charge has

received less attention. Here we investigate the limit of the enhancement (increase) in the coagulation sink of charged particles in the atmosphere. We show that this enhancement limit is asymptotically 2, that is the coagulation sink of charged particle is double that of the otherwise same neutral particles.

   Small charged particles are intrinsically out of equilibrium (Gonser et al., 2014; Hoppel and Frick, 1986; Hõrrak et al., 2008; López-Yglesias and Flagan, 2013a, b; Gopalakrishnan et al., 2013). The true (thermal) equilibrium charge distribution

on particles of diameter $d_{\mathrm{p}}$ is

$$p_i^{\mathrm{eq}}(d_{\mathrm{p}}) = \mathcal{N} \exp\left(-\frac{e^2}{4\pi\epsilon_0\,k_{\mathrm{B}}\,T}\frac{i^2}{d_{\mathrm{p}}}\right) = \sqrt{\frac{1}{2\pi}\frac{d_{\mathrm{C}}}{d_{\mathrm{p}}}}\exp\left(-\frac{i^2}{2}\frac{d_{\mathrm{C}}}{d_{\mathrm{p}}}\right) \tag{1}$$





where $k_\mathrm{B}$ is the Boltzmann constant, $e$ is the elementary charge, $i$ is the charge on the particles, and $\epsilon_0$ is the permittivity of free space. The first quotient in eq. 1 is a scale length—the Coulomb diameter shown in eq. 2.

$$d_\mathrm{C} = \frac{e^2}{2\pi\,\epsilon_0\,k_\mathrm{C}\,T} \qquad (2)$$

At 300 K, $d_\mathrm{C} = 111.4$ nm. For particles smaller than the Coulomb diameter, the energy of even a single elementary charge is well above the thermal energy. This means that for $d_\mathrm{p} \ll d_\mathrm{C}$, any significant charging is far away from equilibrium. It also means that there are two critical sizes for collisions of oppositely charged particles: actual contact, when charge reduction (neutralization) formally occurs, but also passage to within $d_\mathrm{C}$, when charge reduction is viable (López-Yglesias and Flagan, 2013a, b; Gopalakrishnan and Hogan, 2012; Ouyang et al., 2012; Chahl and Gopalakrishnan, 2019). There are thus two possible

rate-limiting events (and a pressure dependence due to third-body collisions within the Coulomb threshold is expected). Even considering relatively inefficient diffusion neutralization by primary ions (Mahfouz and Donahue, 2020), the steady state charged fraction for particles smaller than 7 nm in diameter is extremely small (this is why standard scanning particle sizers are ineffective below that diameter).

For particles larger than roughly 10 nm, the dominant mechanism for gaining and losing charge (in the atmosphere) is

diffusion charging from primary ions (or other sub 10 nm particles if these represent a large fraction of "primary" ions). For $d_\mathrm{p} < d_\mathrm{C}$, most of the particles are neutral, yet the rate of particle neutralization must be balanced by diffusion charging. Thus the collision rate of ions with the (relatively rare by number) charged fraction must equal the collision rate of ions with the (dominant) neutral particles, and the overall collision rate of small charged particles with larger particles will be double that of corresponding neutral particles. Relatively small particles are also exceptionally mobile. This is why the coagulation loss of

small charged particles can be double that of small neutral particles. As shown in Sections 2 and 3 below, this limiting behavior holds only when the background particles are significantly smaller than $d_\mathrm{C}$; it may occur frequently in experiments, and cannot be neglected in the atmosphere, especially in the troposphere.

## 2 Analytic derivation of the limiting behavior for charge coagulation enhancement

We present a simple derivation of this limiting behavior (the doubling of coagulation losses for charged particles). For particles

with $d_p < 100$ nm, there are only three relevant charge states $(-, 0, +)$ with the particles either singly charged or neutral. Given a collision coefficient, $\tilde{\beta}$, and a charge enhancement $\alpha$ for opposite charges, for this derivation only we assume $\tilde{\beta}_{\pm,\mp} = \alpha\,\tilde{\beta}_{0,\mp} = \alpha\,\tilde{\beta}_{\pm,0}$. Likewise, we also assume $\tilde{\beta}_{\pm,\pm} = \gamma\,\tilde{\beta}_{0,\pm} = \gamma\,\tilde{\beta}_{\pm,0}$, where $\gamma$ is the reduction factor for charges of the same sign. Here, $\tilde{\beta}_0 = \tilde{\beta}_{0,\pm} = \tilde{\beta}_{\pm,0} = \tilde{\beta}_{0,0}$. Because all particles are at most singly charged, this applies to bigger particles comprising the coagulation sink, $N_{\mathrm{CoagS},\{-,0,+\}}$, as well as smaller particles potentially lost to coagulation, $N_{\{-,0,+\}}$. We assume that the

bigger particles are such that $N_{\mathrm{CoagS},-} = N_{\mathrm{CoagS},+} = N_{\mathrm{CoagS},\pm}$. When the "diffusion charging" rates of smaller particles to bigger particles are in equilibrium,

$$\tilde{\beta}_{0,\pm}\,N_{\mathrm{CoagS},0}\,N_\pm = R_\pm \quad = \quad R_0 = \tilde{\beta}_{\mp,\pm}\,N_{\mathrm{CoagS},\mp}\,N_\pm \qquad (3)$$





and so, $N_{\mathrm{CoagS},0} = \alpha N_{\mathrm{CoagS},\mp}$. Then, the coagulation sink for neutral particles is

$$
\begin{aligned}
\mathrm{CoagS}_0 &= \tilde{\beta}_0 \left(N_{\mathrm{CoagS},0} + N_{\mathrm{CoagS},+} + N_{\mathrm{CoagS},-}\right) \\
&= \tilde{\beta}_0 \left(N_{\mathrm{CoagS},0} + 2N_{\mathrm{CoagS},\pm}\right) = \tilde{\beta}_0 N_{\mathrm{CoagS},0} \left(1 + 2/\alpha\right)
\end{aligned}
\tag{4}
$$

and for charged particles,

$$
\begin{aligned}
\mathrm{CoagS}_\pm &= \tilde{\beta}_{0,\pm} N_{\mathrm{CoagS},0} + \tilde{\beta}_{\mp,\pm} N_{\mathrm{CoagS},\mp} + \tilde{\beta}_{\pm,\pm} N_{\mathrm{CoagS},\pm} \tag{5} \\
&= \tilde{\beta}_0 N_{\mathrm{CoagS},0} + \alpha \tilde{\beta}_0 N_{\mathrm{CoagS},\mp} + \gamma \tilde{\beta}_0 N_{\mathrm{CoagS},\pm} = \tilde{\beta}_0 N_{\mathrm{CoagS},0} \left(2 + \gamma/\alpha\right) \tag{6}
\end{aligned}
$$

and finally,

$$
\frac{\mathrm{CoagS}_\pm}{\mathrm{CoagS}_0} = \frac{(2 + \gamma/\alpha)}{(1 + 2/\alpha)}
\tag{7}
$$

and so $\mathrm{CoagS}_\pm/\mathrm{CoagS}_0 \to 2$ as $\alpha \gg 2$ and $\gamma \ll 1$. This is the limit when the presence of the charge significantly increases or decreases the collision of particles. The other limit is: $\mathrm{CoagS}_\pm/\mathrm{CoagS}_0 = 1$ as $\alpha = \gamma = 1$, when the presence of charge is insignificant (charge is "screened"). This shows that in the limit of a coagulation sink comprised of relatively small particles (that is, first limit: $\alpha \gg 2$ and $\gamma \ll 1$) , coagulation can be greatly enhanced for small charged particles compared to neutral particles of the same size.

## 3  Computed static limit

To illustrate this derived limit further, we use primary ions as a limit for the smallest particles. We compute the limiting behavior from available data, assuming that the coagulation between smaller particles and bigger particles is the same as the coagulation of primary ions with bigger particles. In this case, we study the coagulation sink ratio, defined by $\mathrm{CoagS}_i/\mathrm{CoagS}_0$ for $i = \pm 1$. Further,

$$
\mathrm{CoagS}_i = \int\limits_{\infty > d_{\mathrm{p}} \geq d_{\mathrm{p}*}} \sum_j \beta_{i,j}\left(d_{\mathrm{p}*}, d_{\mathrm{p}}\right) n_j\left(d_{\mathrm{p}}\right) \mathrm{d}d_{\mathrm{p}},
\tag{8}
$$

where $d_{\mathrm{p}*}$ is the diameter whence particles are formed, $\beta$ is the particle–particle coagulation kernel, and $n$ is the particle number density or size distribution. For ease, we take $n$ as monodisperse distributions, thereby simplifying eq. 8 by dropping the integration.

We assume the smallest particles (at $d_{\mathrm{p}*}$) have characteristics similar to those of primary ions found in the atmosphere. To this end, we utilize the ion–particle attachment coefficients (for $\beta$) and the corresponding charge fraction distributions as reported by López-Yglesias and Flagan (2013a, b). The ion–particle attachment coefficients are shown in Figure 1. We note that López-Yglesias and Flagan (2013a) do not report the case where a neutral small particle is colliding with a bigger particle, akin to a neutral "ion" colliding with a bigger particle, and so we have extrapolated that an acceptable form is similar to the average of positive and negative ions' attachment coefficients to a neutral particle. In Figure 2, we show the coagulation sink

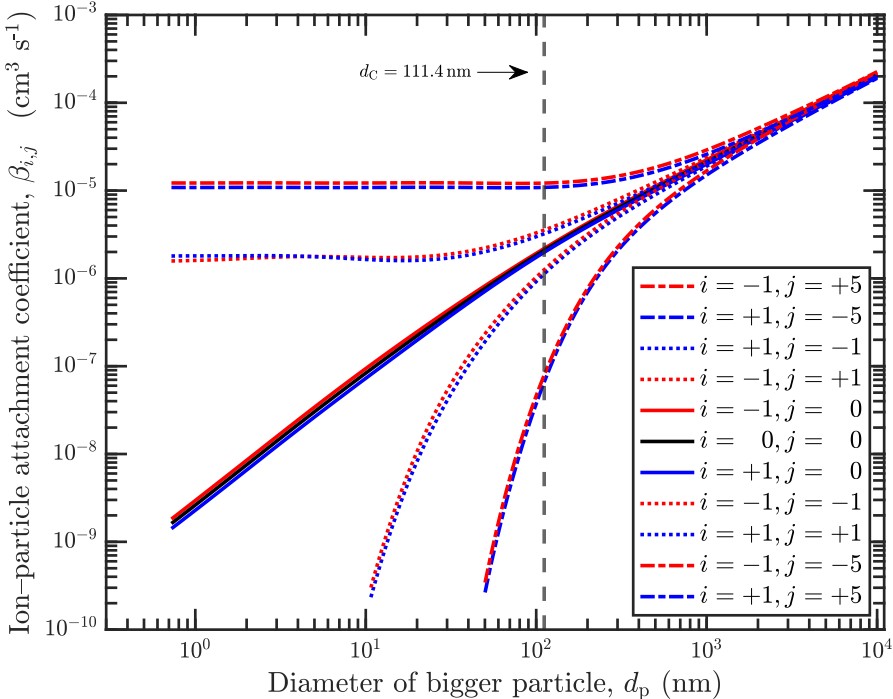

**Figure 1.** Ion–particle flux attachment coefficient kernel of primary ion with charge $i$ and a particle of size $d_\mathrm{p}$. The colors refer to the charge of the smaller particles (modeled as primary ions): red is negative, blue is positive, and black is neutral. The line styles refer to the charge of the bigger particle ($j$): dashed-dotted is $\pm 5$, dotted is $\pm 1$, and continuous is 0. Also shown is the Coulomb diameter, $d_\mathrm{C}$, at 111.4 nm. Neutralization coefficients (attachment between particles of opposite charge) tend toward an asymptotic value for $d_p < d_\mathrm{C}$.

ratio converging onto exactly 2 for both negative and positive when the size of the bigger particles decreases to around 10 nm. And for particles bigger than 100 nm, we show that the coagulation sink ratios are also converging exactly onto 1. They tend to diverge slightly as the particle sizes grows much bigger because of the charging asymmetry between positive and negative ions observed in the atmosphere.

For convenience, we present two parametirzations that capture this limiting behavior in Figure 3. The first parameterization is based an exponential function and takes the form $1 + \exp\left(-(0.025\, d_\mathrm{p})^{1.1}\right)$. The second parameterization is based on the limiting behavior presented earlier. We observe that as $\alpha \gg 1$, then by definition $\gamma \ll 1$. As such, the ratio $\gamma/\alpha$ decreases faster than $1/\alpha$. The relationship between $\alpha$ and $\gamma$ is not simply reciprocal, $\gamma \neq 1/\alpha$. But this is at most a caveat expressed in the $k$th dependency in $\left(2 + 1/\alpha^k\right)/\left(1 + 2/\alpha\right)$, where $1 \leq k \leq 3$ for most cases by observation. What remains is the functional form of

$\alpha$ in size, $\alpha(d_\mathrm{p})$. This functional form of $\alpha$ depends weakly on the number of charges and the size is the leading factor. From observation, $\alpha \approx 1 + 500\, d_\mathrm{p}^{-1.5}$ for $i = \pm 1$ and $\alpha \approx 1 + 5000\, d_\mathrm{p}^{-1.5}$ for $i = \pm 5$.



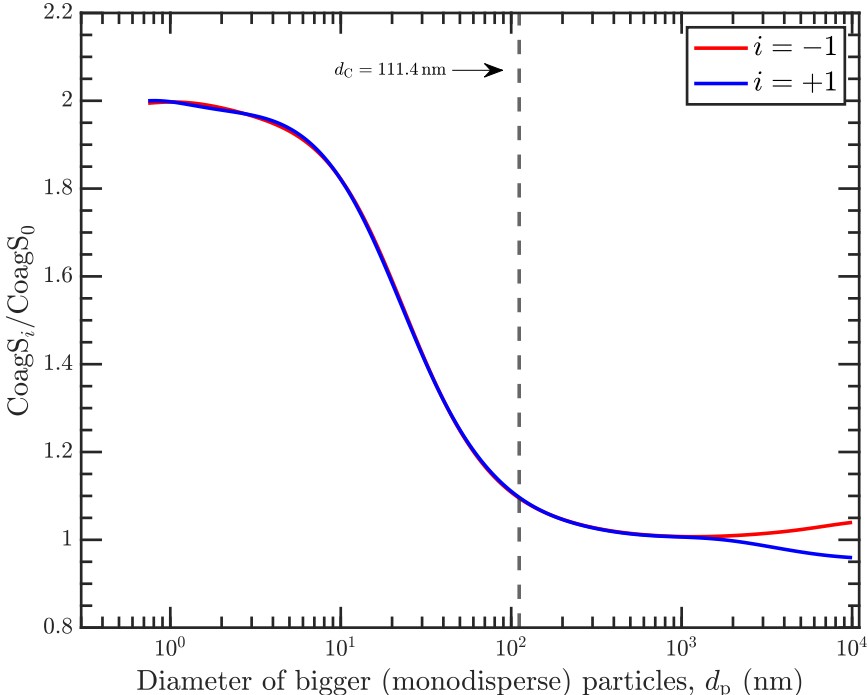

**Figure 2.** The ratio of the coagulation sink of charged ($i = \pm1$) to neutral nucleating particles (modeled as primary ions). The colors refer to the charge of the primary ions: red is negative and blue is positive. Also shown is the Coulomb diameter, $d_C$, at 111.4 nm. For $d_p \ll d_C$, the coagulation sink for charged particles approaches a limit of twice the coagulation sink for neutral particles.

## 4 Conclusions

We have shown that the limit of the enhancement of coagulation scavenging of charged particles is 2 (double that of neutrals). Particle survival probabilities decrease exponentially as the coagulation sink increases (Kulmala et al., 2017), and so all being

equal, the doubling of the coagulation sink can amount to a noticeable drop in the survival probability. In other words, if the survival probability of neutral particles is $p$, then the survival probability of charged particles is $p^2$ where $p \leq 1$. Thus, it is imperative to consider this counterbalancing (blunting) effect when studying ion-induced new-particle formation and ion-enhanced new-particle growth in the atmosphere.

*Author contributions.* NMD conceived of the research question herein; NGAM and NMD conducted the research, interpreted the results,
and wrote the manuscript together.





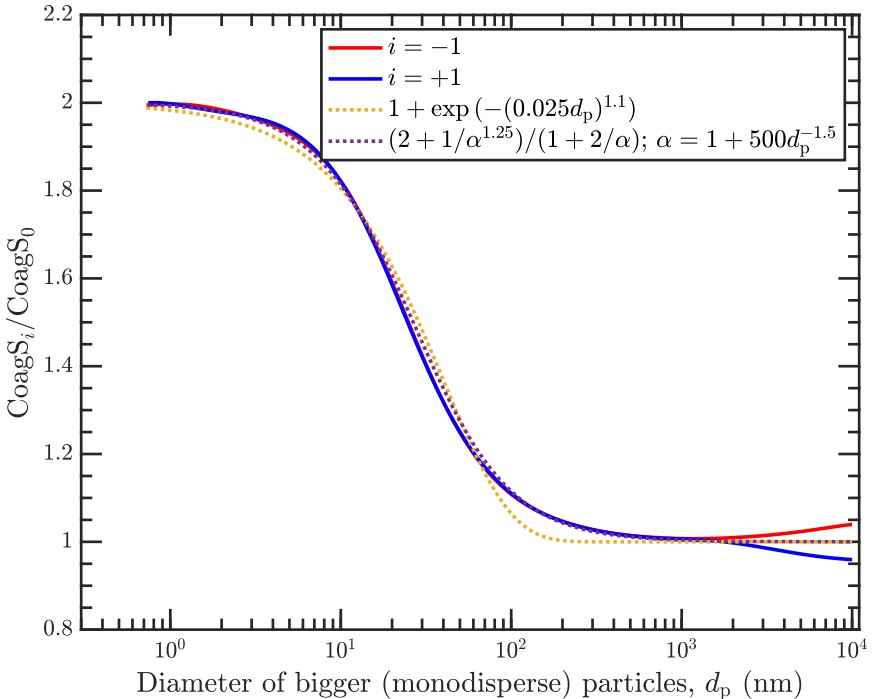

**Figure 3.** Coagulation sink ratios shown in Figure 2, along with two parameterizations capturing the limiting behavior.

*Competing interests.* We declare no competing interests.

*Acknowledgements.* This is a theoretical modeling study: we used no new data data. Parts of this work were supported by the National Science Foundation (NSF) under Grants AGS1740665 and AGS1801897.



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
