# Peer review of "Technical note: the enhancement limit of coagulation scavenging of small charged particles"

_Atmospheric Chemistry and Physics, 2020_

## Referee Comment (RC1) · Anonymous Referee #1 · 20 Nov 2020

**Review of 'Technical note: the enhancement limit of coagulation scavenging of small charged particles'**

**General comments**

In this study, the effect of particle charge on coagulation is studied theoretically. More specifically, it is shown that the enhancement of coagulation of charged particles has an upper limit of 2. The authors note that this effect should be considered, when studying atmospheric new particle formation involving ions.

The derivation of the coagulation enhancement limit seems sound, and the result can be useful when analyzing measurements or modelling new particle formation in the presence of charge. However, some references to previous research should be added, and some parts need to be clarified. Also, more discussion on the implications of the results is needed. I recommend the article for a publication after the authors have considered these revisions.

**Specific comments**

Line 5: In the abstract and conclusions it is mentioned that this effect is important when studying ion-induced new particle formation and ion-enhanced particle growth. However, these processes and their importance in the atmosphere (or in laboratory experiments) are not discussed elsewhere in the manuscript. In the introduction, the study is motivated more generally by the exponential dependence of particle survival probability on coagulation sink. Although this is true, if ions are generally unimportant in new particle formation, the enhancement of coagulation scavenging due to charge is also unimportant. Therefore, some discussion on particle formation and growth involving ions should be added to the introduction.

Line 14-15: In the statement "the role of charge has received less attention" it is not entirely clear if the authors are referring to the role of charge more generally or in some specific issue. Generally, the current knowledge of the effect of charge on coagulation should be clarified in the introduction. Now there are no references to articles on coagulation in the presence of charge, although relevant studies seem to exist (for example Ghosh et al. 2017 and references therein).

Line 32-33: A reference to a study showing the small charged fraction is needed.

Line 34-35: A reference is needed here (related to main charging mechanism).

Line 45: A reference is needed here (related to the charging state of sub-100 nm particles).

Line 39: Can you clarify how the fact that small particles are mobile is connected to the higher coagulation rate of small charged particles compared to small neutral particle?

Line 61-62: It is unclear to me how this limit also corresponds to the case where the presence of charge decreases collisions.

Line 63-64: Can you clarify why the first limit corresponds to the case where coagulation sink comprises of relatively small particles? Although it is shown in the next section, it is not clear here how this conclusion can be made.

Line 77-80: Can you clarify if the results from López-Yglesias and Flagan are based on some laboratory experiments? How do they correspond to other estimations for ion-aerosol attachment coefficients, for example those in Horrak et al. (2008)? And how does your extrapolation for neutral particles correspond to commonly used expressions for neutral particle collision rates?

Line 93-98: It would be good to add some discussion here on the conditions in which this limit of enhancement of coagulation is reached, and on possible practical applications where this result can be used. Finally, when discussing atmospheric implications of the result, the importance of ions in new particle formation could be clarified (see my first specific comment).

**Technical comments**

Eq. (1): All the symbols used in the equations should be explained, for example here it is unclear what the first symbol on the right-hand side refer to.

Line 34-42: It would be good to try to avoid using parenthesis excessively (here and elsewhere in the manuscript).

Line 46-48: Please clarify what different subscripts in collision coefficients etc. refer to.

Eq. (8): Here you could also clarify what subscript $i$ and $j$ refer to.

Line 80: You could start a new paragraph here before "In Figure 2,.."

Figure 1: Please refer in the figure caption to the article from where the coefficients are taken.

Line 85: There is a typo here (parametrizations).

Figure 3: Please explain in the figure caption what different lines refer to.

**References**

Ghosh, K., Tripathi, S. N., Joshi, M., Mayya, Y. S., Khan, A., & Sapra, B. K. (2017). Modeling studies on coagulation of charged particles and comparison with experiments. *Journal of Aerosol Science*, *105*, 35-47.

Horrak, U., Aalto, P. P., Salm, J., Komsaare, K., Tammet, H., Mäkelä, J. M., ... & Kulmala, M. (2008). Variation and balance of positive air ion concentrations in a boreal forest. *Atmos. Chem. Phys.*, 8, 655–675,

---

## Referee Comment (RC2) · Anonymous Referee #2 · 12 Jan 2021

This manuscript is very difficult to read because of missing references, especially related with assumptions used, and insufficient explanation of symbols and subscripts. The topic itself and especially the final result is definitely interesting and useful, if generally true. For readers to properly follow the theoretical treatment, a more thorough writing is, however, needed in my opinion.

Some remarks:

Equation 1: Please add a reference and explain what the variable N means.

Equation 2. Reference?

[Figure]

Lines 32-37: This explanation in words of particle dynamics would be easier to follow if supported by equations.

Equations 3 – 7 and discussion leading to them: Please explain all variables and subscripts. Especially, what is the difference between beta, beta_(0,0), beta_(0,plusminus), beta_(0,minusplus), beta_(plusminus,0), beta_(minusplus,0), beta_(plusminus,minusplus) and beta_(minusplus,plusminus)? Also, what is the difference between N, N_{-,0,+}, N_(CoagS,0), N_(CoagS,+), N_(CoagS,-), N_(CoagS, plusminus) and N_(CoagS, minusplus)?

Based on equations 3-7, the background aerosol seems to be assumed monodisperse. Is this true? If yes, mention this clearly.

Lines 46-47: It is mentioned that also particles that form the coagulation sink can only be singly charged. Is this true? Reference?

Lines 45 and 48 and equation 3: It is assumed that the collision processes are symmetrical with respect to sign. Furthermore, if I understand correctly, collisions between charged and neutral are assumed to be occurring with same rate as collisions between neutrals. How well is this generally true? References?

Equation 3: The theory would be easier to follow if the dynamical equations describing the dynamics of the particle concentrations would be written first and the equilibrium equations (which are not trivial for the general reader) thereafter.

Equation 7: Is this equation valid generally or only with the assumptions made, i.e. e.g. charge symmetry and monodispersity of the coagulation sink?
* * *

---

## Author Response (AR1)

**Author comments on the reviews of "Technical note: the enhancement limit of coagulation scavenging of small charged particles"**

We thank the two anonymous referees for their time and feedback in reviewing this technical note. We have edited this paper based on their feedback to better the readability and accessibility of the writing and presentation. We have added additional references to support our statements throughout. Here is our detailed response in black to their comments in blue.

**1. Referee #1**

In this study, the effect of particle charge on coagulation is studied theoretically. More specifically, it is shown that the enhancement of coagulation of charged particles has an upper limit of 2. The authors note that this effect should be considered, when studying atmospheric new particle formation involving ions. The derivation of the coagulation enhancement limit seems sound, and the result can be useful when analyzing measurements or modelling new particle formation in the presence of charge. However, some references to previous research should be added, and some parts need to be clarified. Also, more discussion on the implications of the results is needed. I recommend the article for a publication after the authors have considered these revisions.

As a general comment, we emphasize that this is a Technical note, and thus a relatively limited exploration of the implications is intentional. However, proper referencing, clarity are certainly essential, and we improved each in our revised note.

**1.1. Specific comments**

- Line 5: In the abstract and conclusions it is mentioned that this effect is important when studying ion-induced new particle formation and ion-enhanced particle growth. However, these processes and their importance in the atmosphere (or in laboratory experiments) are not discussed elsewhere in the manuscript. In the introduction, the study is motivated more generally by the exponential dependence of particle survival probability on coagulation sink. Although this is true, if ions are generally unimportant in new particle formation, the enhancement of coagulation scavenging due to charge is also unimportant. Therefore, some discussion on particle formation and growth involving ions should be added to the introduction.

    We have added a discussion about the role of ions in particle formation and growth, stressing the reported evidence of enhancement of formation rates[6,11] and noting that the effect on the growth rate is negligible once particles are above 2 nm.[8]

- Lines 14–15: In the statement "the role of charge has received less attention" it is not entirely clear if the authors are referring to the role of charge more generally or in some specific issue. Generally, the current knowledge of the effect of charge on coagulation should be clarified in the introduction. Now there are no references to articles on coagulation in the presence of charge, although relevant studies seem to exist (for example Ghosh et al. [2] and references therein).

    Following our response above, we explain this statement more clearly in the text as follows. Without charge, the role of the counterbalancing between growth and coagulation scavenging has received attention.[7] Charge can enhance new particle formation[6,11] and, to a lesser extent, growth.[8] The counterbalancing between growth and coagulation scavenging *in the presence of charge* has received less attention. We focus on one aspect of this; namely, the coagulation scavenging enhancement due to charge. **We study the effect**

[Figure]

Figure 1: The steady-state fraction of particle carrying charges $i$ as a function of size $d_\mathrm{p}$. Below 10 nm, there are hardly any charged particles at all (less than 10% of singly charged particles). Below 100 nm, almost all particles are neutral or singly charged (less than 5% are doubly charged or more).

**of charge on the coagulation sink and we conclude that, in the limit, the presence of charge can amount to double of the coagulation sink, which consequently amounts to squaring of the survival probability. We also provide an easy expression for this behavior that could be used in modeling studies as a limit on the coagulation sink dependency on charge.** We agree with Referee #1 that there are studies calculating coagulation in the presence of charge and we do cite a few of them.[1,9,10] Our goal, however, is not to calculate the coagulation rate, per se, but the coagulation sink, which is more pertinent to the survival probability. That is, the goal of this paper is not to introduce a new way of calculating coagulation in the presence of charge, but to use existing methodology of calculating coagulation in the presence of charge to illuminate the potential effects of charge on the survival probability.

- Lines 32–33: A reference to a study showing the small charged fraction is needed.

  We have added a reference.[9] Note though that any particle charge distribution—and hence any study thereof—will show that the charged fraction of small particles is extremely small. As an example, see Figure 1 which shows the fraction of particles carrying charges $i$ as a function of size, $d_\mathrm{p}$. This figure is reproduced based on data by López-Yglesias and Flagan [9].

- Line 45: A reference is needed here (related to the charging state of sub-100 nm particles).

  We again cite the same paper.[9] Note the preceding response and the associated figure.

- Line 39: Can you clarify how the fact that small particles are mobile is connected to the higher coagulation rate of small charged particles compared to small neutral particle?

  We feel that adding this is outside the scope of the technical note; an easy way to think

about it is that both forces—Brownian and Coulombic—are inversely proportional to mass. The interested reader can find plenty of references cited to learn more. Since this is outside the scope of this paper, we rephrased the sentences here to unlink them—essentially, deleting "why"—and so hopefully, this will not become a distraction from the ideas presented.

- Lines 61–62: It is unclear to me how this limit also corresponds to the case where the presence of charge decreases collisions.

    The correspondence applies by definition (effectively detailed balance). The preceding statement lays out the conditions $\alpha \gg 2$ (meaning significant increase in the like–unlike case) and $\gamma \ll 1$ (meaning significant decrease in the like–like case). The "presence of charge" is only half the story; the other half is whether the coagulating particles have the same charges (like–like) or opposing charges (like–unlike). We have added some clarification about this in the main text. Note that if one of the particles is neutral (does not have a charge), we assume the presence of charge makes no difference then. This is brought up later by both Referees, and yes, it is a reasonable assumption.

- Lines 63–64: Can you clarify why the first limit corresponds to the case where coagulation sink comprises of relatively small particles? Although it is shown in the next section, it is not clear here how this conclusion can be made.

    We added a clarifying statement that the only case when $\alpha \gg 2$ and $\gamma \ll 1$ is when the particles in the coagulation sink are quite small, as shown in the figures later on.

- Lines 77–80: Can you clarify if the results from López-Yglesias and Flagan [9, 10] are based on some laboratory experiments? How do they correspond to other estimations for ion–aerosol attachment coefficients, for example those in Hõrrak et al. [5]? And how does your extrapolation for neutral particles correspond to commonly used expressions for neutral particle collision rates?

    The López-Yglesias and Flagan [9, 10] studies are not based on laboratory experiments; they are modeling studies. They are slightly more accurate than those of Hõrrak et al. [5], which are based almost entirely on Hoppel and Frick [4]. They are slightly more accurate because they take into account higher-order dependencies like the ones related to charged–neutral coagulation. However, for our specific application here, using any of the above will lead to identical conclusions. As for our convention of simply averaging the neutral–positive and neutral–negative collisions to get the neutral–neutral collisions, we did that so that we obtain all of our parameters from the López-Yglesias and Flagan [9, 10] studies—nothing else. This ensures self consistency. They result in extremely similar data points to the ones calculated by commonly used expressions. Moreover, the reason López-Yglesias and Flagan [9, 10] studies deviate ever so slightly from the commonly used expressions when calculating rates involving a neutral particle, is the higher-order dependencies. At the end of the day, it is important that modelers use the same source for the calculations involving steady state charged fractions, ion–particle fluxes, and particle–particle fluxes. Otherwise, there is a real danger in having the dynamics not converge correctly; that is, there is a danger in the dynamic calculations via fluxes yielding different states than those directly calculated from the steady-state equations if they come from different sources. Again though, this issue would not affect the conclusions presented here—not even the slightest.

- Lines 93–98: It would be good to add some discussion here on the conditions in which this limit of enhancement of coagulation is reached, and on possible practical applications where this result can be used. Finally, when discussing atmospheric implications of the result, the importance of ions in new particle formation could be clarified (see my first specific comment).

We added a brief discussion of ion-induced new-particle formation and ion-enhanced new-particle growth earlier in the introduction. Here we add the following sentence, "We note that ion-induced new-particle formation and ion-enhanced new-particle growth only happen if there is an abundance of ions (and therefore charges) available."

**1.2. Technical comments**

- Eq. (1): All the symbols used in the equations should be explained, for example here it is unclear what the first symbol on the right-hand side refer to.

  Everything is directly defined except the first symbol. The first symbol is a normalization factor, which is actually indirectly defined by the equality sign.

- Lines 34–42:It would be good to try to avoid using parenthesis excessively (here and elsewhere in the manuscript).

  We agree parentheses are not pretty and we have therefore eliminated most of them for a variety of other punctuation marks.

- Lines 46–48: Please clarify what different subscripts in collision coefficients etc. refer to.

  Done.

- Eq. (8): Here you could also clarify what subscript $i$ and $j$ refer to.

  Done.

- Line 80: You could start a new paragraph here before "In Figure 2,.."

  Sure!

- Figure 1: Please refer in the figure caption to the article from where the coefficients are taken.

  Done.

- Line 85: There is a typo here (parametrizations).

  Fixed!

- Figure 3: Please explain in the figure caption what different lines refer to.

  The lines are explained in the previous figure and the point of this figure is to draw attention to the parameterizations which are shown in the legend and mentioned in the caption.

**2. Referee #2**

This manuscript is very difficult to read because of missing references, especially related with assumptions used, and insufficient explanation of symbols and subscripts. The topic itself and especially the final result is definitely interesting and useful, if generally true. For readers to properly follow the theoretical treatment, a more thorough writing is, however, needed in my opinion.

We appreciate the perspective of fresh eyes and apologize for the lack of clarity. We have addressed these concerns by rewriting portions of the paper, adding references, and expanding on ideas. We made sure that all symbols and subscripts are defined, and in some cases we have expanded compact representations to improve readability.

**Some remarks**

- Equation 1: Please add a reference and explain what the variable N means.

  This can be derived readily from first principles; it is just an equilibrium distribution, but we have added an oft-cited paper by Gunn and Woessner [3]—maybe the first to state this formulation accounting for the different negative and positive mobilities of atmospheric ions. In our equation, we assume the mobilities are identical for negative and positive ions. The variable $N$ (now $A$) is a normalization factor of the probability distribution.

- Equation 2. Reference?

  To our knowledge this length scale has not previously been named nor has it been discussed in the literature. We think it deserves its own name, and so we suggest the Coulomb diameter. It is an important length scale and we hope the (now-edited) following paragraph illustrates why it is important.

- Lines 32–37: This explanation in words of particle dynamics would be easier to follow if supported by equations.

  We agree! However, our goal is to give the reader a very quick summary of the complex issues surrounding the coagulation of charged particles. We have edited these lines for clarity and added a few additional references to guide the reader.

- Equations 3–7 and discussion leading to them: Please explain all variables and subscripts. Especially, what is the difference between $\beta$, $\beta_{0,0}$, $\beta_{0,\pm}$, $\beta_{0,\mp}$, $\beta_{\pm,0}$, $\beta_{\mp,0}$, $\beta_{\pm,\mp}$ and $\beta_{\mp,\pm}$? Also, what is the difference between $N$, $N_{-,0,+}$, $N_{CoagS,0}$, $N_{CoagS,+}$, $N_{CoagS,-}$, $N_{CoagS,\pm}$ and $N_{CoagS,\mp}$?

  Our mistake. For brevity, we sacrificed clarity. We fixed this by clearly defining all these terms and breaking the discussion paragraph into two. We also made the derivation (equations) into their own separate paragraph so to speak.

- Based on equations 3–7, the background aerosol seems to be assumed monodisperse. Is this true? If yes, mention this clearly.

  Yes, the background aerosol in the derivation of the limiting behavior is monodisperse, which we now state explicitly.

- Lines 46–47: It is mentioned that also particles that form the coagulation sink can only be singly charged. Is this true? Reference?

  We assume this only for the limiting case, and we now emphasize this in the text. Note we are concerned with the small-particle (coagulation sink) limit here. So, one could take that limit to be under 100 nm, where this assumption would be quite reasonable. See Figure 1 which could be reproduced from available data.[4,9]

- Lines 45 and 48 and equation 3: It is assumed that the collision processes are symmetrical with respect to sign. Furthermore, if I understand correctly, collisions between charged and neutral are assumed to be occurring with same rate as collisions between neutrals. How well is this generally true? References?

  The answer to both questions is "yes"; they are both assumptions. The first assumption is somewhat straightforward—the order of charges should not make a difference. The second difference is a little trickier, but it is also quite reasonable. The López-Yglesias and Flagan [9, 10] studies do include higher-order dependencies that partly include some of the issues arising in the neutral–charged collisions. But overall, the effect of these higher-order dependencies is minimal and it will affect our conclusions here. So, it is safe to assume the the neutral–charged and neutral–neutral collisions are the same, at least in the context

of this study. In fact, it is easily verifiable; the neutral–charged data will be quite similar if calculated either way (using our averaging or using commonly used expressions). We have made a conscious decision to use all our data from the same source[9,10] because, as we explain in our response above, there is a real danger in using different sources for different aspects of the calculations.

One could include those higher-order dependencies, such as atmospheric particles having fundamentally different mobilities for different charges due to the different chemical compositions of positive and negative atmospheric ions. In fact, Figure 1 (in this response) does show the resulting different steady states of the particle charged fractions. We are confident that these differences will not impact the results here though, especially in light of the fact, in the computed section (3), we do use results from studies that include higher-order dependencies. That in the end is why this is a technical note and not a full paper; in other work we intend to make use of this general finding about the limiting behavior of the survival probability, but in those applications we will use a more general form of the coagulation kernel.

- Equation 3: The theory would be easier to follow if the dynamical equations describing the dynamics of the particle concentrations would be written first and the equilibrium equations (which are not trivial for the general reader) thereafter.

  We agree, but then again, our goal is not to give a reader a thorough treatment of the dynamics here; instead, a quick summary that is appropriate for this type of paper: a technical note.

- Equation 7: Is this equation valid generally or only with the assumptions made, i.e. e.g. charge symmetry and monodispersity of the coagulation sink?

  Strictly speaking, this equation is only valid for the assumptions made. However, you can see later on (e.g. Figure 3 in the paper) that it is quite good even when the assumptions are relaxed. Indeed, we have more work showing this is a really good equation and these results apply if the assumptions are relaxed even further. This is precisely why we wanted this brief technical note as a standalone published work—without the noise of more complex dynamics.

**Correspondence:** Neil M. Donahue (nmd@andrew.cmu.edu)

**Abstract.** We show that the limit of the enhancement of coagulation scavenging of charged particles is 2, that is, doubled compared to the neutral case. Because the particle survival probability decreases exponentially as the coagulation sink increases, all else being equal, the doubling of the coagulation sink can amount to a dramatic drop in survival probability—squaring the survival probability, $p^2$, where $p \leq 1$ is the survival probability in the neutral case. Thus, it is imperative to consider this counterbalancing effect when studying ion-induced new-particle formation and ion-enhanced new-particle growth in the atmosphere.

**1 Introduction**

There are many situations in atmospheric phenomena where we care about total particle number, and especially total particle number above some critical size. An example is cloud activation, where the total number of cloud condensation nuclei is often estimated to be the total number of particles with diameters  $d_{\mathrm{p}} \geq 50$ nm ($\mathcal{N}_{50}$) or perhaps  $d_{\mathrm{p}} \geq 100$ nm ($\mathcal{N}_{100}$), depending on updraft velocity (Rosenfeld et al., 2019; Pierce and Adams, 2007; Pierce et al., 2012; Gordon et al., 2017; Lee et al., 2019). When  new-particle formation, also known as nucleation, or emission is dominated by much smaller particles, we  care not only about the formation or emission rate but also the survival probability of  the newly formed particles as they grow to the critical size.

The particle survival probability decreases exponentially as the coagulation sink increases (Kulmala et al., 2017; Kerminen and Kulmala, 2002; Lehtinen et al., 2007; Li and McMurry, 2018). The counterbalancing role of growth rate and coagulation sink is well known,  for example as studied by Kulmala et al. (2017). Additionally, the presence of charge can increase new-particle formation rates in both acid-base (Merikanto et al., 2016) and organic (Kirkby et al., 2016) systems. Charge can also increase the growth rate of small particles due to the polar enhancement of gas–particle collision parameters (Lehtipalo et al., 2018), though this effect tends to be negligible once particles are bigger than 2 nm. Yet, the direct role of charge in the counterbalancing of growth rate and coagulation sink has received less attention. Here, we focus on the effect of charge on the coagulation sink—we investigate the limit of the enhancement  in the coagulation sink of charged

particles that can take place in the atmosphere or in experiments. We show that this enhancement limit is asymptotically 2; that is, the coagulation sink of charged particle is double that of the otherwise same neutral particles.

Small charged particles are intrinsically out of equilibrium (Gonser et al., 2014; Hoppel and Frick, 1986; Hõrrak et al., 2008; López-Yglesias and Flagan, 2013a, b; Gopalakrishnan et al., 2013). The thermal equilibrium charge distribution on particles of diameter $d_\mathrm{p}$ is given via the probability density function, $p_i^\mathrm{eq}$, assuming positive and negative ion mobilities are the same; see the similar expression provided by Gunn and Woessner (1956) for different mobilities.

$$p_i^\mathrm{eq}(d_\mathrm{p}) = \mathcal{N}\mathcal{A} \exp\left(-\frac{e^2}{4\pi\epsilon_0\,k_\mathrm{B}\,T}\frac{i^2}{d_\mathrm{p}}\right) = \sqrt{\frac{1}{2\pi}\frac{d_\mathrm{C}}{d_\mathrm{p}}}\exp\left(-\frac{i^2}{2}\frac{d_\mathrm{C}}{d_\mathrm{p}}\right) \tag{1}$$

In eq. 1, $\mathcal{A}$ is normalization factor of the distribution, $k_\mathrm{B}$ is the Boltzmann constant, $e$ is the elementary charge, $i$ is the charge on the particles, and $\epsilon_0$ is the permittivity of free space. The first quotient in eq. 1 is a scale length—which we call the Coulomb diameter and is shown in eq. 2.

$$d_\mathrm{C} = \frac{e^2}{2\pi\,\epsilon_0\,k_\mathrm{CB}\,T} \tag{2}$$

At 300 K, $d_\mathrm{C} = 111.4$ nm. For particles smaller than the Coulomb diameter, the energy of even a single elementary charge is well above the thermal energy. This means that for $d_\mathrm{p} \ll d_\mathrm{C}$, any significant charging is far away from equilibrium. It also means that there are two critical sizes for collisions of oppositely charged particles: actual contact, when charge reduction (neutralization) formally occurs, but also passage to within $d_\mathrm{C}$, when charge reduction is viable (López-Yglesias and Flagan, 2013a, b; Gopalakrishnan and Hogan, 2012; Ouyang et al., 2012; Chahl and Gopalakrishnan, 2019). There are thus two possible rate-limiting events, in addition to an expected pressure dependence due to third-body collisions within the Coulomb threshold. Even considering relatively inefficient diffusion neutralization by primary ions (Mahfouz and Donahue, 2020), the steady state charged fraction for particles smaller than 7 nm in diameter is extremely small (López-Yglesias and Flagan, 2013a); relatedly, this is why standard scanning particle sizers are ineffective below this diameter.

For particles larger than roughly 10 nm, the dominant mechanism for gaining and losing charge (in the atmosphere) is diffusion charging, either from primary ions or other sub 10 nm particles if these represent a large fraction of extremely small and mobile ions. For $d_\mathrm{p} < d_\mathrm{C}$, most of the particles are neutral—see for example the studies by López-Yglesias and Flagan (2013a) and Hoppel and Frick (1986)—and yet at steady state the rate of particle neutralization must be balanced by diffusion charging. Thus, the collision rate of ions with the (relatively rare by number) charged fraction must equal the collision rate of ions with the (dominant) neutral particles, and the overall collision rate of small charged particles with larger particles will be double that of corresponding neutral particles. Relatively small particles are also exceptionally mobile. We show that the coagulation loss of said small charged particles can be double that of small neutral particles. As shown in Sections 2 and 3 below, this limiting behavior holds only when the background particles are significantly smaller than $d_\mathrm{C}$; it may occur frequently in experiments, and cannot be neglected in the atmosphere, especially in the troposphere.

**2 Analytic derivation of the limiting behavior for charge coagulation enhancement**

We present a simple derivation of this limiting behavior, where the presence of charge leads to the doubling of coagulation losses. For particles with $d_p < 100$ nm, there are only three relevant charge states $(-, 0, +)$ with  particles either singly charged or neutral.; the fraction of particles with two or more charges is truly negligible (López-Yglesias and Flagan, 2013a). We assume a collision coefficient, $\tilde{\beta}$, and a charge enhancement $\alpha$ for opposite charges; for this derivation only, we assume $\tilde{\beta}_{\pm,\mp} = \alpha\tilde{\beta}_{0,\mp} = \alpha\tilde{\beta}_{\pm,0}$, where the subscripts on $\tilde{\beta}$ refer to the charge state of the coagulating particles. That is, $\tilde{\beta}_{\pm,\mp}$ means $\tilde{\beta}$ when the first particle has a positive charge and the second a negative one or the first negative and the second positive—in other words, the coagulating particles have opposing charges. Likewise, we also assume $\tilde{\beta}_{\pm,\pm} = \gamma\tilde{\beta}_{0,\pm} = \gamma\tilde{\beta}_{\pm,0}$, where $\gamma$ is the reduction factor for charges of the same sign.; like before, $\tilde{\beta}_{\pm,\pm}$ means $\tilde{\beta}$ of two particles carrying the same charge. Additionally, we define $\tilde{\beta}_0 = \tilde{\beta}_{0,\pm} = \tilde{\beta}_{\pm,0} = \tilde{\beta}_{0,0}$; that is, we drop the two subscripts for one when the coagulation involves a neutral particle as we assume a neutral–neutral collision rate is the same as neutral–charged collision rate. All of this is to say: we assume that the order of charges do not matter; $\alpha$ is the enhancement factor due to like–unlike coagulation; $\gamma$ is the suppression factor due to the like–like coagulation; the neutral–neutral coagulation is the same as the neutral–charged coagulation; all particles have at most one charge.

Because all particles are at most singly charged in this limiting derivation, this applies to bigger particles comprising the coagulation sink, $N_{\mathrm{CoagS},\{-,0,+\}}$, as well as smaller particles potentially lost to coagulation, $N_{\{-,0,+\}}$. That is, $N_{\mathrm{CoagS},-}$ is the number of monodisperse particles in the coagulation sink (bigger particles) which have a negative charge and $N_-$ is the number of monodisperse newly formed (or smaller) particles which have a negative charge. We assume that  positive and negative mobilities of smaller particles, including primary ions or newly formed particles, are the same; and as such, the number of positive and negative bigger particles is the same—that is, $N_{\mathrm{CoagS},-} = N_{\mathrm{CoagS},+} = N_{\mathrm{CoagS},\pm}$.

 We assume the "diffusion charging"  rates—$R_{\{-,0,+\}}$—of smaller particles to bigger particles are in equilibrium such that $R_- = R_+ = R_0$. We write $R_k$ as the rate of ions or smaller particles, $N_{\{-,0,+\}}$, coagulating with bigger particles, $N_{\mathrm{CoagS},\{-,0,+\}}$, to form a particle of charge $k$. Here, $k$ is $-1$,

$$\tilde{\beta}_{0,\pm} N_{\mathrm{CoagS},0} N_{\pm} = R_{\pm} \quad = \quad R_0 = \tilde{\beta}_{\mp,\pm} N_{\mathrm{CoagS},\mp} N_{\pm}$$

 0, or $+1$, and so without confusion, we call those states $-$, $0$, and $+$. We assume the negative and positive mobilities are the same; in the atmosphere, they are different, but this assumption will help us realize the limit. As such, we use $R_{\pm}$ to mean $R_-$ or $R_+$ like before; note that if the subscript $\pm$ appears alongside $\mp$ in the equations below, it means the charges are opposite.

Accordingly, we define the "diffusion charging" rates and set them in equilibrium in eq. 3.

[revised manuscript text omitted]

We assume the smallest particles (at $d_{\mathrm{p}*}$) have characteristics similar to those of primary ions found in the atmosphere. To this end, we utilize the ion–particle attachment coefficients (for $\beta$) and the corresponding charge fraction distributions as reported by López-Yglesias and Flagan (2013a, b). The ion–particle attachment coefficients are shown in Figure 1. We note that López-Yglesias and Flagan (2013a) do not report the case where a neutral  extremely small "particle" is colliding with a bigger particle, akin to a neutral "ion" colliding with a bigger particle, and so we have extrapolated that an acceptable form is similar to the average of positive and negative ions' attachment coefficients to a neutral particle. This averaging leads to similar results found elsewhere for the neutral–neutral attachment coefficients of particles those sizes. We opted to use this averaging as opposed to commonly used expression to ensure we use all data below from the same source.

In Figure 2, we show the coagulation sink ratio converging onto exactly 2 for both negative and positive when the size of the bigger particles  (comprising the coagulation sink) is less than around 10 nm. And for particles bigger than 100

[revised manuscript text omitted]